# Hydrodynamic manipulation of nano-objects by optically induced thermo-osmotic flows

Martin Fränzl [1] & Frank Cichos [1✉]

Manipulation of nano-objects at the microscale is of great technological importance for constructing new functional materials, manipulating tiny amounts of fluids, reconfiguring sensor systems, or detecting tiny concentrations of analytes in medical screening. Here, we show that hydrodynamic boundary flows enable the trapping and manipulation of nano-objects near surfaces. We trigger thermo-osmotic flows by modulating the van der Waals and double layer interactions at a gold-liquid interface with optically generated local temperature fields. The hydrodynamic flows, attractive van der Waals and repulsive double layer forces acting on the suspended nanoparticles enable precise nanoparticle positioning and guidance. A rapid multiplexing of flow fields permits the parallel manipulation of many nano-objects and the generation of complex flow fields. Our findings have direct implications for the field of plasmonic nanotweezers and other thermo-plasmonic trapping systems, paving the way for nanoscopic manipulation with boundary flows.

[1] Peter Debye Institute for Soft Matter Physics, Molecular Nanophotonics Group, Universität Leipzig, Linnéstr. 5, 04103 Leipzig, Germany.
✉email: cichos@physik.uni-leipzig.de

The control and manipulation of nano-objects is a key element for future nanophotonics[1–5], material science[4,6,7], biotechnology[2,8,9], or even quantum sensing[10]. Analytes dissolved in liquids, for example, need to be delivered, concentrated, separated, or locally confined for further studies to become eventually processed and removed. Photonic elements including plasmonic nanostructures require precise positioning or controlled rearrangements to serve as adaptive functional structures. Key elements of the control at the micro- and nanoscale are often either pressure-driven fluidics transporting liquid volume and solutes or the generation of potential energy landscapes or force fields. The latter is achieved with optical[3] and plasmonic tweezers[11,12], magnetic fields[13], or using electrokinetic[14] or opto-electronic[15] effects. Especially in the field of plasmonic tweezers and nanoantennas, where light is used to excite collective electron motion in noble-metals, the Joule losses lead to the unavoidable generation of heat at boundaries as an unwanted side effect[16,17]. Yet, such optically generated temperature fields seem also suitable for the manipulation of nano-objects in liquids, for example, for the trapping of nanoparticles[18] and single molecules[19] or protein aggregates[20] as well as for manufacturing active particles[21–24]. Those techniques rely on a drift of molecules and particles in optically generated temperature gradients termed thermophoresis or suggest thermo-electric effects[25] relying on a thermally induced charge separation. In addition, thermo-electrohydrodynamic effects using time-varying electric fields have been proposed for rapid particle transport[26,27] and convective effects that arise from temperature-induced density changes in the large liquid cells have been reported[28–31].

Here we report on a fundamental physical process that is able to provide versatile trapping and manipulation of nano-objects and fluids near surfaces in the simplest geometries. Contrary to most other techniques, our scheme is based on hydrodynamic flows generated by optically induced thermo-osmosis. Thermo-osmosis relies on a perturbation of the interfacial interactions at a solid–liquid boundary and is present in all experiments involving temperature gradients in plasmonic structures including plasmonic tweezers. We show that local temperature gradients on a thin gold film induce strong interfacial flows of several 10–100 µm s$^{-1}$ in its direct vicinity (10 nm) that results in a flow pattern reminiscent of convection. Based on a fully quantitative analysis of our experimental results we reveal that these thermo-osmotic flows on gold–water interfaces are induced by a temperature-induced perturbation of the van der Waals (vdW) interactions. Nano-objects suspended in the liquid are, therefore, dragged by the hydrodynamic forces originating from these flows. Utilizing attractive vdW interactions of the nano-object with the gold surface or temperature-induced depletion, we trap and manipulate different types of nano-objects near the surface. The fast heating at small scales allows us to multiplex flow fields and to manipulate multiple objects with great precision. Our detailed analysis of the flow fields, the localization accuracy of nano-objects, and a comparison with numerical and theoretical predictions provide a quantitative understanding of these effects and paves the way for controlling boundary layer dynamics to manipulate objects at the smallest length scales in solutions.

## Results and discussion

**Experimental configuration and working principle.** Our experiments rely on a simple sample geometry with a gold film (50 nm) that is deposited on a microscopy glass coverslip (Fig. 1a). The sample chamber contains a suspension of gold nanoparticles (AuNPs) or other nano-objects (polystyrene NPs and ellipsoids) with a controlled amount of salt (NaCl), surfactants (SDS, ...), or polymers (PEG). The gold film is heated locally

in an inverted microscopy setup by a highly focused laser (532 nm) using beam steering optics (Acusto-Optic Deflector, AOD). The nano-objects are observed using darkfield illumination with an oil-immersion darkfield condenser (NA 1.2) and a ×100 oil-immersion objective set to NA 0.6 (Fig. 1b). Additional details of the experimental setup and sample preparation are provided in the Methods section, Supplementary Note 1 and Fig. S1.

The trapping of nano-objects as detailed in the following is comprising two effects. (i) The vertical confinement of the suspended objects as achieved by an attractive interaction of the suspended nano-objects with the gold surface, which is found to be the vdW interaction for gold nanoparticles and can be replaced by depletion forces for other materials. (ii) The generation of thermo-osmotic boundary flows that are induced by the local heating and the corresponding perturbation of the liquid–solid interactions. This boundary flow is directed radially inwards to the heated spot and provides a confining hydrodynamic force on suspended objects at the heating spot.

**Dynamics of AuNPs close to a Au film.** Consider a single AuNP with a radius of $R = 125$ nm that is suspended in an aqueous solution of NaCl at a concentration of $c_0 = 10$ mM and diffusing in a thin liquid film of about 3 µm thickness over a 50 nm Au film (Fig. 1a). Exploring the diffusion of the particle we observe a restriction of the $z$-positions to a thin layer close to the gold film (see Supplementary Notes 2, 3 with Figs. S2, S3 for details). The gold particle never defocuses under these conditions while it does in deionized (DI) water (Supplementary Video 1). This restricted out-of-plane motion is the result of interactions comprising an attractive vdW contribution, a repulsion of the electrostatic double layers of the particle and surface[32] as described by the DLVO theory and the gravitational potential (see Supplementary Notes 4, 5 for details):

$$V(d, c_0) = V_{\mathrm{E}}(d, c_0) + V_{\mathrm{vdW}}(d) + V_{\mathrm{G}}(d). \quad (1)$$

The total potential for the experimental situation is depicted in Fig. 2a for different salt concentrations (Figs. S4–S8 and Supplementary Note 4 for parameters) as a function of the surface-to-surface distance $d = z - R$, where $z$ denotes the distance of the particle center from the surface. The stronger screening of the surface charges at the gold film and the AuNP at higher salt concentration increase the importance of attractive vdW interactions to create this secondary minimum in the DLVO part of the potential. This potential influences the observed dynamics and leads also to a stronger hydrodynamic coupling of the particle to the nearby gold surface[33]. The in-plane $D_\parallel$, Eq. (2), and out-of-plane $D_\perp$ diffusion coefficient (see Supplementary Note 6 and Figs. S11, S12 for details) are modulated with the distance $z$ of the particle from the wall:

$$\frac{D_\parallel(z)}{D_0} \approx 1 - \frac{9}{16}\frac{R}{z} + \frac{1}{8}\left(\frac{R}{z}\right)^3 \pm \cdots := \gamma_\parallel^{-1}(z). \quad (2)$$

Over the course of a diffusion trajectory, the particle samples different regions with different diffusion coefficients according to its probability density $p(d) \propto \exp(-V(d, c_0)/(k_{\mathrm{B}}T))$ to be at a distance $d$ from the surface (filled regions in Fig. 2a). The observed in-plane diffusion coefficient is, thus, a weighted average of the diffusion coefficient over the different vertical positions $d$. Using $p(d)$ we can calculate the corresponding salt concentration dependence of the in-plane diffusion coefficient and compare that to the experimental results. Figure 2b shows that the experimentally observed $D_\parallel$ is decreasing with increasing salt concentration due to the hydrodynamic coupling in fair agreement with the theoretical predictions (for three different Hamaker

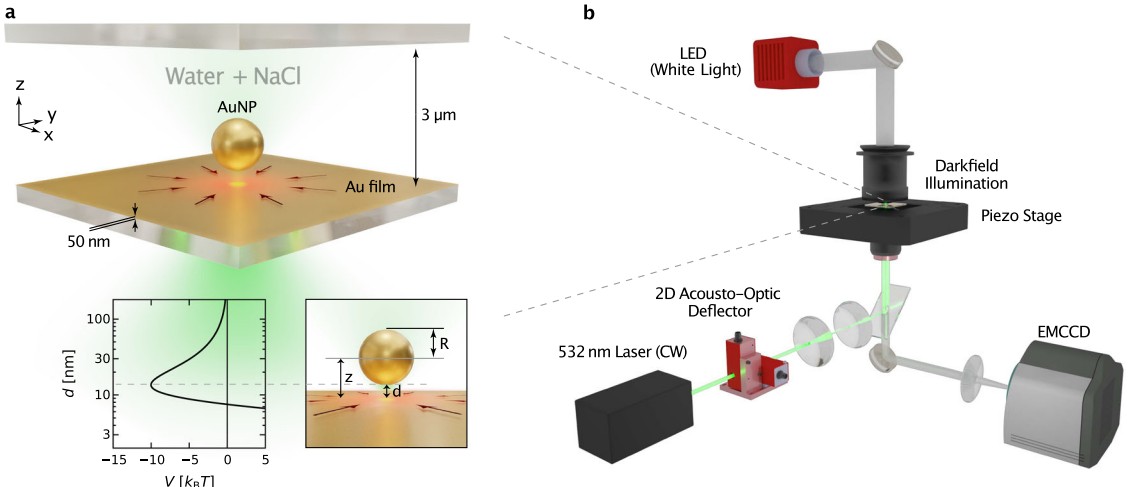

**Fig. 1 Thermo-hydrodynamic manipulation of AuNPs in NaCl solution. a** The sample consists of two glass slides that confine a 3 μm thin liquid film of gold nanoparticles (AuNPs) with radius $R$ dispersed in aqueous NaCl solution. The lower glass slide carries a 50 nm Au film that is locally heated by optical absorption of a focused laser of $\lambda = 532$ nm wavelength. The inset depicts the DLVO (Derjaguin, Landau, Verwey, Overbeek) potential as described in the main text and Supplementary Notes 4 for a $R = 125$ nm AuNP in 10 mM NaCl. **b** The experimental setup comprises an inverted optical microscope equipped with a steerable focused laser of $\lambda = 532$ nm wavelength controlled by an acusto-optic deflector. The AuNPs are observed using darkfield illumination with an oil-immersion darkfield condenser (NA 1.2) and a ×100 oil-immersion objective set to NA 0.6. Images are recorded with an EMCCD camera, typically, with an inverse framerate of $\tau = 20$ ms.

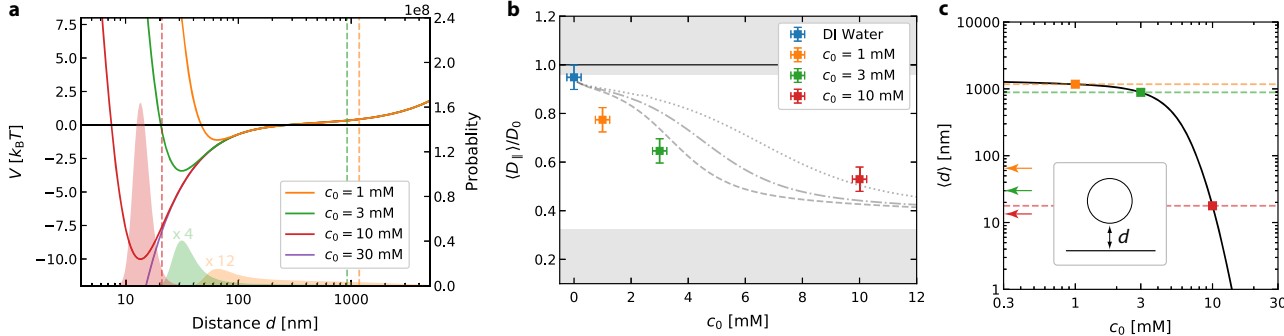

**Fig. 2 DLVO potential and lateral diffusion analysis. a** Plot of the DLVO potential, Eq. (1), between a 250 nm AuNP and a 50 nm Au film on a glass surface as function of surface–surface distance $d$ for different NaCl concentrations $c_0$. The shaded curves display the calculated probability density for finding the particle at this distance at the different salt concentrations (see Supplementary Notes 4 for details). The vertical dashed lines correspond to the mean distance of the particle as calculated from the probability density for a 3 μm liquid film height. **b** The measured diffusion coefficient $\langle D_\parallel \rangle / D_0$ parallel to the Au film with respect to the bulk diffusion coefficient $D_0$ as function of the NaCl concentration $c_0$. Symbols correspond to the experimental values measured without additional laser heating of the gold film. The lines reflect the theoretical prediction including a distance dependent diffusion coefficient for three different Hamaker constants (dotted: $A_H = 4 \times 10^{-20}$ J, dash-dotted: $5 \times 10^{-20}$ J, and dashed: $6 \times 10^{-20}$ J) of gold according to a Boltzmann weighting (see main text). The error bars have been estimated from pipetting errors and errors from the MSD fitting. **c** Relation between the mean distance $\langle d \rangle$ and the NaCl concentration $c_0$. The symbols and the horizontal lines denote the calculated distances for measured concentrations. The arrows indicate the secondary minimum of the corresponding DLVO potentials.

constants for the AuNP gold surface interaction). If one additionally calculates the mean distance $\langle d \rangle$ of the particle from the surface using $p(d)$, it can be seen that the minimum of the DLVO potential significantly affects the mean distance only for $c_0 > 3$ mM. For $c_0 = 10$ mM, the potential minimum corresponds approximately to the mean distance $\langle d \rangle$, indicating that the particle is completely trapped near the DLVO potential minimum and is not delocalized over the entire film thickness as at lower concentrations. Using the calculated mean distance as a function of concentration $c_0$, it is also possible to estimate the mean distance $\langle d \rangle$ of the particles from the surface in the experiments, which is about 1.5 μm and 0.9 μm at the lowest NaCl concentrations (Fig. 2c). At a concentration of $c_0 = 10$ mM the particle is hovering at a distance of $\langle d \rangle = 20$ nm surface. Note that this corresponds to values of $z/(2R) \approx 0.58$, which is far below the

commonly explored region of the hydrodynamic coupling of colloids to walls[33] allowing to experimentally explore new terrains also in the field of hydrodynamic wall coupling of colloids.

**Hydrodynamic particle confinement**. When tightly focusing the light of 532 nm wavelength to the gold film, a part of the incident energy (about 30%) is absorbed and converted into heat that perturbs the liquid–solid interactions. The temperature rise at the gold surface can be determined using a thin nematic liquid crystal (5CB) film and substantiated by finite element simulations with the complete three-dimensional temperature profile in the solution (see Fig. 3a, b and Supplementary Notes 7, 8, Figs. S13–18 for details).

These local temperature perturbations of the solid–liquid interactions at the interface induce a thermo-osmotic flow[34,35]

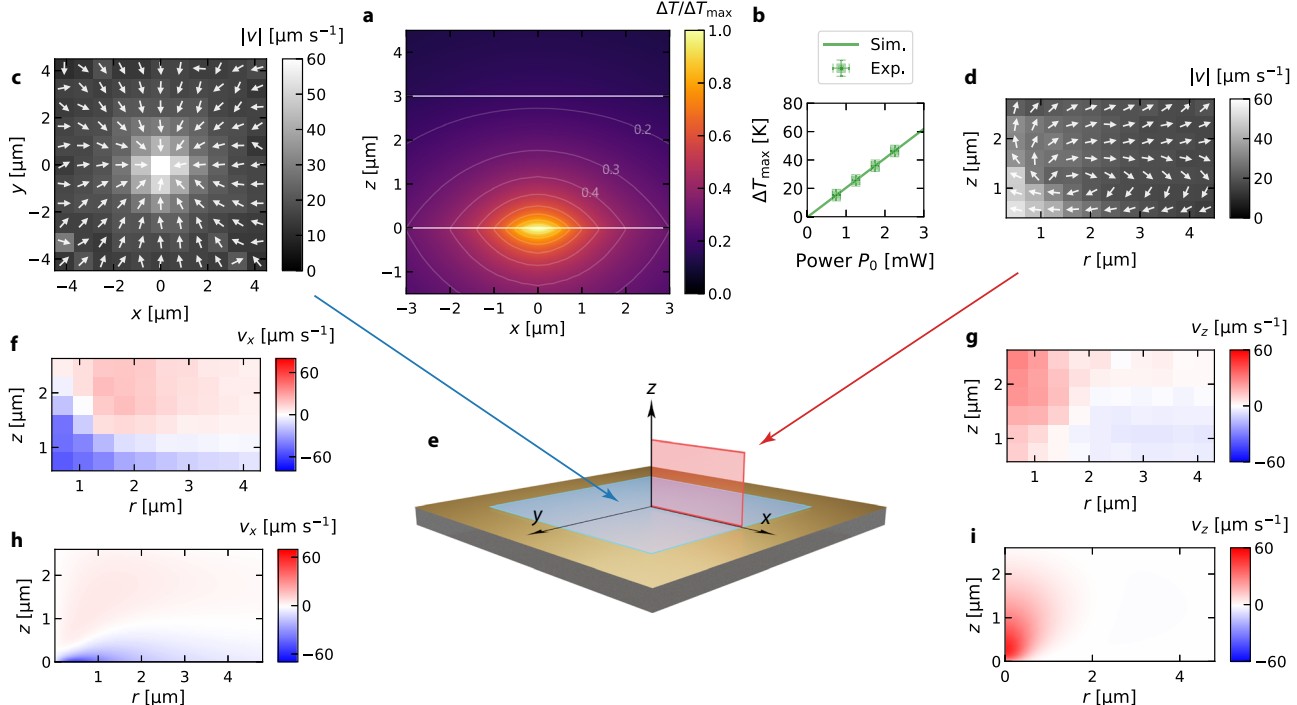

**Fig. 3 Temperature distribution and thermo-osmotic flow field. a** Relative temperature increment $\Delta T/\Delta T_{max}$ in the $xz$-plane of the sample as obtained from numerical simulations. The simulation is detailed in Supplementary Note 7. The white lines correspond to the gold and glass interface, respectively. **b** Experimentally obtained temperature increment $\Delta T_{max}$ as a function of the incident laser power $P_0$ (green data points) compared to the simulated values (green curve). The experimental data have been obtained with the help of a liquid crystal as explained in Supplementary Note 8. The error bars have been obtained from uncertainties of the power measurement and the estimation of phase transition radius. **c** Measured thermo-osmotic flow field in the $xy$-plane in close proximity to the gold film ($z < 500$ nm) for a laser power of 2 mW ($\Delta T_{max} = 40$ K) for a NaCl concentration $c_0 = 0$ mM (DI water). The grayscale image indicates the magnitude of the velocity. The arrows display the direction of the flow. **d** Measured thermo-osmotic flow field in the $xz$-plane for the same parameters as in **c**. The grayscale image indicates the magnitude of the velocity. The arrows display the direction of the flow. **e** Illustration of the measured flow field planes in **c** and **d**. **f, g** The $x$- and $z$-component of the measured flow velocities for the same parameters as in **c** and **d** compared to the simulation results in **h** and **i**.

(Supplementary Note 9, Figs. S19–S22). Taking a liquid volume element close to the solid from the cold side and exchanging that with one on the hot side would not only transport heat since the liquid volumes have different temperatures, but also additional free energy as the liquid has a different interaction with the solid in these regions. The flow is induced in an ultrathin boundary layer corresponding in thickness to the length scale of liquid–solid interactions. Since the characteristic interaction length of liquid–solid interactions is only a few nanometers, the boundary flow on the substrate can be collapsed into a quasi-slip hydrodynamic boundary condition:

$$v_\parallel = -\frac{1}{\eta}\int_0^\infty z\,h(z)\,\mathrm{d}z\,\frac{\nabla_\parallel T}{T} = \chi\,\frac{\nabla_\parallel T}{T}\,, \qquad (3)$$

where $\eta$ is the viscosity of the liquid, $h(z)$ is the excess enthalpy, $T$ the temperature and $\nabla_\parallel T$ is the temperature gradient parallel to the surface. The integral can be summarized to a thermo-osmotic coefficient $\chi$. The thermo-osmotic coefficient $\chi$, therefore, contains all information about the interfacial interaction between the liquid and the solid. If $\chi < 0$ the liquid is driven to the cold, whereas for $\chi > 0$, the liquid is driven to the hot. These boundary flows are present at all liquid–solid interfaces with tangential temperature gradients, though, they are commonly overlooked. They become particularly important for plasmonic and thermo-plasmonic trapping[16], as those techniques rely on the dynamics of molecules and particles in the direct vicinity of plasmonic nanostructures. The boundary flow drives the flow field inside the fluid film. The resulting volumetric flow field can be tracked

experimentally by single AuNPs in DI water, where the particles are not confined to a surface layer as reported in the previous section. We analyze the in-plane ($xy$) position of the particle and its $z$-position, where the latter is estimated from the radius $r_0$ of the defocussed particle images (see Supplementary Video 2 and Supplementary Note 2 for details). The measured velocity distributions in the $xy$-plane near the gold layer and in the $xz$-plane are shown in Fig. 3c and d, respectively. The $x$- and $z$-component of the measured flow velocities are depicted in Fig. 3f, g. The flow profiles compare well to the simulation results shown in Fig. 3h, i, although some discrepanices exist especially close to $r = 0$. Particle velocities are highest in this region and can easily reach several 10 µm s$^{-1}$, which increases the particle localization errors for finite exposure time and finally blurs the velocity profile additionally. From the velocity measurements, we extract a thermo-osmotic coefficient on the order of $\chi \sim 10 \times 10^{-10}$ m$^2$ s$^{-1}$ (see Supplementary Note 9 for details). We can break down the contributions to this value with Eqs. (4) and (5) to estimate the double layer and vdW contributions using the experimental parameters. Note that AuNPs do not show thermophoresis due to their high thermal conductivity and thus isothermal surface (see Supplementary Note 10 for details)

$$\chi_E = \frac{\varepsilon\zeta^2}{8\eta} \approx 0.8 \cdot 10^{-10}\ \mathrm{m^2\,s^{-1}}\,. \qquad (4)$$

For the electrostatic contribution we used a zeta potential of $\zeta = -30$ mV[36] and a static permittivity of $\varepsilon = 80\,\varepsilon_0$. An estimate

of the vdW contribution can be given by:

$$\chi_{\mathrm{vdW}} = \frac{A_{\mathrm{H}}\beta T}{3\pi\eta d_0} \approx 9.3 \cdot 10^{-10}\ \mathrm{m^2\ s^{-1}}, \qquad (5)$$

in terms of the Hamaker constant $A_{\mathrm{H}}$ between water and the Au film with $\beta = 0.2 \times 10^{-3}\,\mathrm{K^{-1}}$ being the thermal expansion coefficient of water and $d_0 = 0.2$ nm for the cut-off parameter[34] (see Supplementary Note 9 for details). The sum of both contributions $\chi = \chi_{\mathrm{E}} + \chi_{\mathrm{vdW}} = 10.1 \times 10^{-10}\,\mathrm{m^2\ s^{-1}}$ matches well the experimental result and suggests that thermo-osmosis at gold–water interfaces is governed by vdW interactions. The obtained quasi-slip velocities are ranging up to 80 μm s⁻¹ and provide, due to their omnipresence, a unique tool for nanofluidics. These thermo-osmotic flows are induced without any external pressure difference. They can be controlled by the light intensity of the heating laser and are quickly switched due to the extremely fast heat conduction at these length scales. Moreover, the finding of the vdW dominated thermo-osmotic flows suggest that such contributions must be present in any plasmonic trapping experiment with extended gold structures[12,16,17,27].

Using $F_x^{\mathrm{TF}} = 6\pi\eta R\,\gamma_\parallel\,v_x$ and $F_z^{\mathrm{TF}} = 6\pi\eta R\,\gamma_\perp\,v_z$, where $\gamma_\parallel$ and $\gamma_\perp$ are the correction factors for the friction coefficient of a sphere close to a surface, we are able to extract the hydrodynamic forces that are exerted on the AuNP tracers (see Eq. (2), Fig. S9 and Supplementary Notes 4, 6 for details). The lateral forces allow to confine objects at the heating spot, yet the hydrodynamic force normal to the surface (z-direction) is repulsive without any additional interaction. Finally, such boundary flows with substantial vertical velocity gradients also exhibit vorticity (see Supplementary Note 9, Fig. S23 for details) that generates a torque on suspended objects causing them to rotate[37].

**Single particle trapping and flow field multiplexing.** The vertical hydrodynamic drag force $F_z^{\mathrm{TF}}$ is superimposed with an attractive force due to the DLVO potential when increasing the NaCl concentration. The surface-to-surface distance between AuNP and gold film and the depth of the appearing secondary DLVO potential minimum can be controlled by the NaCl concentration. At a NaCl concentration of about $c_0 = 10$ mM, the attractive potential has a depth of about 10 $k_{\mathrm{B}}T$ (see Fig. 2a) and is strong enough to compete with the vertical drag force and additional optical forces on the AuNP to trap the particle above the heating spot.

Supplementary Video 3 demonstrates this trapping of a AuNP above the hot spot on the Au surface. This is purely the result of the hydrodynamic drag forces generated by the thermo-osmotic flow and the attractive vdW interaction between the AuNP and the Au film. This observation is substantiated by a quantitative evaluation of the lateral trap stiffness and vertical forces, as depicted in Fig. 4a, b. The fluctuations of the particle in the hydrodynamic flow arise from a balance of the restoring hydrodynamic currents and the diffusive currents. Analysis of the lateral position histograms (inset in Fig. 4c for 1.25 mW) yields an effective stiffness of the trap (Fig. 4) that well matches the predictions based on the thermo-osmotic flow (Fig. 3h, i), when converting the lateral flow speeds into forces using the previously mentioned Stokes friction force for $F_x^{\mathrm{TF}}$ (see Supplementary Note 11 and Fig. S27 for details). The hydrodynamic trapping stiffness increases linearly up to a heating power of about 1.8 mW. At this power, the vertical forces become strong enough to let the particle escape the secondary DLVO minimum, which is visible from the z-position time traces displayed in Fig. 4d. The AuNPs are then observed to move vertically out of the DLVO potential to follow the flow inside the sample and to eventually return to the boundary flow via sedimentation (Fig. 4e,

Supplementary Video 4). The forces which eject the particle from the potential comprise the hydrodynamic and the optical forces due to the radiation pressure from the heating laser leaked through the film (see Supplementary Note 12 and Fig. S28 for details). We have evaluated the individual contributions in simulations. They are shown together with the hydrodynamic force and the total vertical force as compared to the attractive force of the DLVO potential (Fig. 4b) and provide quantitative agreement (threshold heating power of 2.25 mW) with our experimental results. Note that while the stationary distribution of particles in the vertical direction is not influenced by the diffusive dynamics, the escape rate from the potential well is heavily altered by the fact that the vertical diffusion coefficient $D_\perp$ of the particle is decreasing to zero when approaching the gold film. This is enhancing the trapping times considerably (see Fig. S10 and Supplementary Note 5 for details) but also increases the time required for the particle to enter the DLVO minimum by diffusion.

The observed trapping is, hence, a vdW assisted thermohydrodynamic process. Vertical confinement is achieved by vdW attraction and double layer repulsion, while lateral confinement is the result of thermo-osmotic flows induced in an ultrathin sheet of liquid at the interface. No additional contributions, for example, due to convective flows with similar flow patterns (see Supplementary Note 13, Figs. S29–S31 for details) or thermoelectric effects are required for a quantitative description[25,31,38,39]. Precise tuning of the DLVO potential enables the trapping of even smaller AuNPs (Fig. 5a, Supplementary Video 5).

The speed of heat diffusion, which is about 4 orders of magnitude faster than the particle diffusion[40] allows us to introduce a flow field multiplexing. We switch the heating location between different positions inducing thermo-osmotic flow fields for time periods of about 100 μs. With the help of this multiplexing, we are able to hold multiple $R = 125$ nm AuNPs (Fig. 5b, c) at distances of less than 1 μm, which would not be possible with continuous heating of close-by locations (Supplementary Videos 6 and 7). A trapped AuNP can also be guided along the predefined path over the Au film as fast as 10 μm s⁻¹ (Fig. 5d, Supplementary Video 8). At larger manipulation speeds ($f > 100$ Hz) and higher heating power ($P_0 > 10$ mW) the thermoosmotic attraction to the heating spot is combined with thermoviscous flows[41,42]. These flows originate from the temperature-dependent viscosity $\eta(T)$ of the liquid and are directed opposite to the scanning direction of the laser[42]. The result of this combination of thermo-osmosis and thermo-viscous flows is a rotating ring-like particle structure (Fig. 5e and Supplementary Video 9).

These different effects that can be exploited in a simple planar geometry give rise to numerous applications including, for example, a freely configurable nanoparticle on mirror geometry for plasmonic sensing[5,43]. The multiplexing of the local flow fields may be helpful to construct more complex effective flow fields for efficient transport of analytes without external pressure.

**Beyond thermo-osmotic van der Waals trapping.** So far, the presented manipulation is based on thermo-osmotic flows that drive the lateral motion of suspended colloids and vertical confinement due to the secondary minimum of the DLVO potential between AuNP and Au film. While the thermo-osmotic flows are characteristic for all systems containing a heated gold–water interface including all previous studies on thermo-plasmonic trapping, the DLVO potential minimum is much weaker for other materials like polymer colloids or macromolecules due to their smaller vdW attraction. Often, those systems even show a repulsion from the heat source due to thermophoresis, which is not

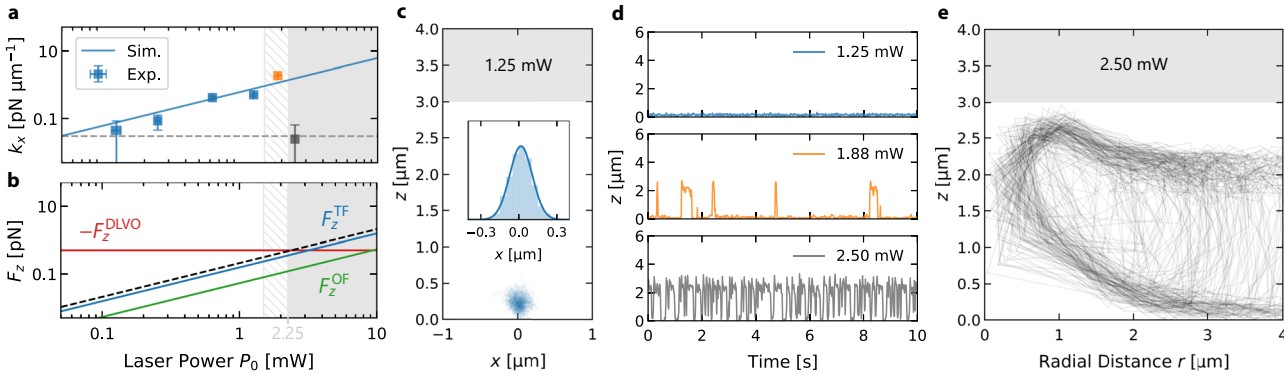

**Fig. 4 Forces on trapped NPs in 10 mM NaCl. a** The lateral trap stiffness $k_x$ obtained from the variance $\sigma_x^2$ of the experimental lateral position histograms as function of the laser power $P_0$ (blue data points, $k_x = k_B T / \sigma_x^2$, see the inset in **c** for the lateral position histogram at $P_0 = 1.25$ mW). The blue solid line represents the simulation result obtained from the lateral velocity field in Fig. 3h (see Supplementary Note 9 for details). Within the dashed area the particle intermittently escapes the trapping potential in vertical direction and transitions into a regime of instant vertical repulsion for higher laser powers (shaded area). The error bars have been estimated from uncertainties of the power measurement and errors from the histogram fitting. **b** The $z$-component of the thermo-osmotic drag force $F_z^{TF}$ (blue line), the optical force $F_z^{OF}$ (green line) and the total force $F_z^{OF} + F_z^{TF}$ (black dashed line) as function the incident laser power $P_0$ for a NP located at $x = 0$, $d = 30$ ($z = d + R$). The attractive DLVO force $F_z^{DLVO}$ is independent of the incident laser power and depicted as horizontal, red line. **c** Trajectory of a AuNP for a heating laser power of 1.25 mW (see Supplementary Video 2 for details). The inset shows the corresponding lateral distribution histogram. **d** Time traces of the $z$-position for three different laser powers. **e** Trajectory of a AuNP for a heating laser power of 2.5 mW, which is above the threshold power of 2.25 mW.

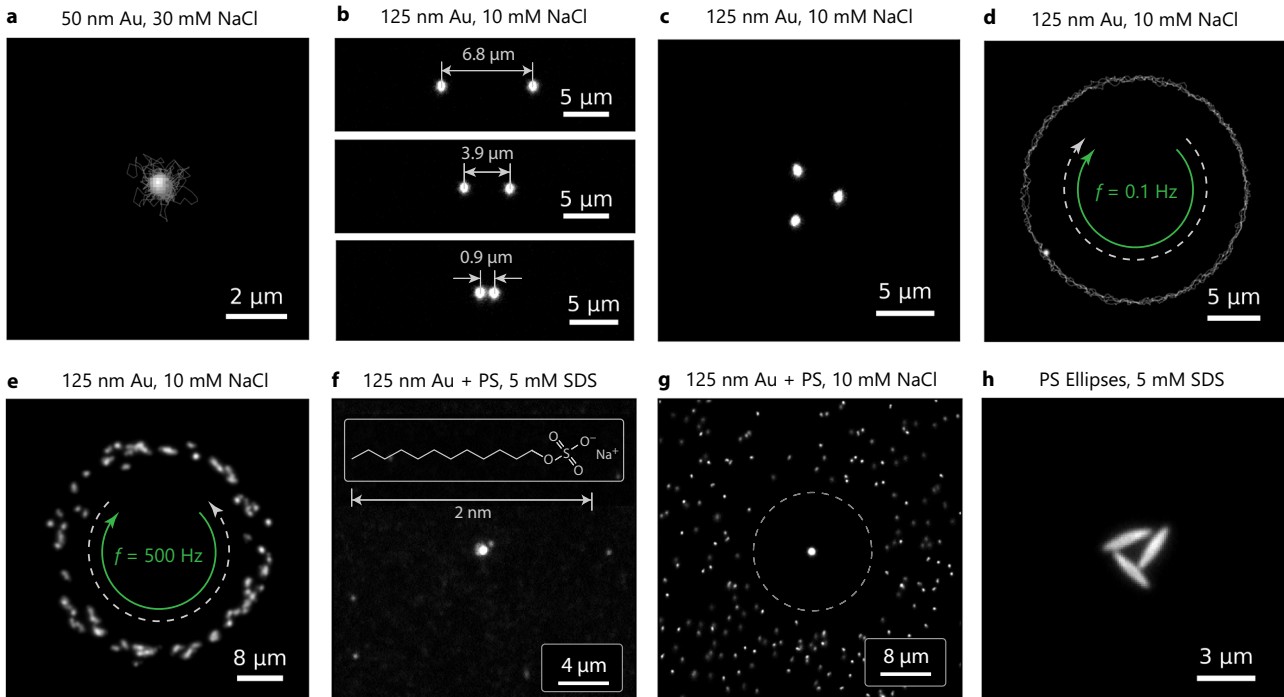

**Fig. 5 Manipulation of NPs over a Au film in NaCl and SDS solution. a** AuNP with 50 nm radius trapped at a NaCl concentration of 30 mM (Supplementary Video 5). **b** Manipulation of two AuNPs by a multiplexed laser beam (Supplementary Video 6). **c** Control of three AuNPs (Supplementary Video 7). **d** Actuation of a single AuNP on a circular trajectory by a steerable laser beam (Supplementary Video 8). The green and white, dashed arrows denote the moving direction of the laser focus and the particle, respectively. **e** Generation of thermo-viscous flows by rotating the laser focus on a circle with a rotation frequency of $f = 500$ Hz at high laser powers (Supplementary Video 9). Note that laser movement (green arrow) and the thermo-viscous flow (white, dashed arrow) and have opposite directions. **f** Attraction of a AuNP and PS NPs in 5 mM SDS due to depletion (Supplementary Video 10). **g** An AuNP (125 nm radius) trapped in an ensemble of polystyrene (PS) NPs of the same size at 10 mM NaCl (Supplementary Video 11), where the PS particles are repelled due to thermophoresis. **h** Attraction of PS ellipsoids (2.39 μm major-axis length, 0.34 μm minor-axis length) in 5 mM SDS (Supplementary Video 12).

present for AuNPs (see Supplementary Note 10, Figs. S24–S26 for details). A more generalized strategy, therefore, needs additional attractive contributions, which confine suspended colloids or molecules to regions close to the gold surface to take advantage of the thermo-osmotic flow.

Such attractive contributions can arise from depletion interactions[21,44]. Thereby a temperature gradient repels dissolved molecules from the heated regions generating a concentration gradient that drives suspended nano-objects to the heating spot. To demonstrate this effect we use the surfactant sodium dodecyl sulfate (SDS) at a concentration of 5 mM well below the critical micelle concentration (8.2 mM) to avoid complications of micelle formation. We suspend additional polystyrene particles (PS) and AuNPs of the same size ($R = 125$ nm) in the solution and compare their dynamics to a solution with AuNPs and PS particles without SDS but 10 mM NaCl. Remarkably, the heated spot is attractive for both AuNPs and for PS NPs (Fig. 5f, Supplementary Video 10) in the SDS solution showing even PS colloidal crystal growth, while only the AuNP is trapped in the NaCl solution and the PS particles are repelled by thermophoresis (Fig. 5g, Supplementary Video 11).

The observations in NaCl are readily explained by the fact that the AuNP is confined in the DLVO minimum as demonstrated above but the PS particle is not due to a 10 times lower Hamaker constant (see Supplementary Note 4 for details). The PS particle samples the whole liquid film thickness equally and not preferentially the region close to the Au film and experiences an additional thermophoretic drift velocity given by:

$$\mathbf{u}_{\mathrm{T}} = -\frac{2}{3}\chi\frac{\nabla T}{T} = -D_{\mathrm{T}}\nabla T,\tag{6}$$

where $D_{\mathrm{T}}$ is the thermophoretic mobility[34] and $\nabla T$ the temperature gradient (Fig. 5g, Supplementary Video 11). For $\chi > 0$ the particle is driven to the cold. From Eq. (4) we find $\chi \approx \chi_{\mathrm{E}} = 1.28 \times 10^{-10}$ m$^2$ s$^{-1}$ and $D_{\mathrm{T}} \approx 0.3$ μm$^2$ K$^{-1}$ s$^{-1}$, where we have used a measured zeta potential of $\zeta \approx -38$ mV (see Methods section for details). The vdW contribution, $\chi_{\mathrm{vdW}}$ to either the thermophoretic drift or the attraction to the gold surface can be neglected due to the smaller Hamaker constant of PS. From the stationary probability distribution of the PS NP we find a Soret coefficient of $S_{\mathrm{T}} \approx 0.24$ K$^{-1}$ in agreement with our theoretical prediction $S_{\mathrm{T}} = D_{\mathrm{T}}/D_{\parallel} \approx 0.21$ K$^{-1}$.

In the SDS solution, the additional surfactant molecules now undergo thermophoresis to yield a concentration gradient in which suspended colloidal particles drift. The lower concentration in the heated regions promotes an effective attractive interaction of suspended colloids with the gold surface due to depletion forces. The total drift velocity $\mathbf{u}$ is then described by[21,34,44].

$$\mathbf{u} = \mathbf{u}_{\mathrm{T}} + \mathbf{u}_{\mathrm{D}} = -\left(D_{\mathrm{T}} - \frac{k_{\mathrm{B}}}{3\eta}R_{\mathrm{m}}^2 c_0 N_{\mathrm{A}}\left(TS_{\mathrm{T}}^{\mathrm{m}} - 1\right)\right)\nabla T,\tag{7}$$

where $\mathbf{u}_{\mathrm{D}}$ denotes the depletion-induced drift velocity (see Supplementary Note 14 and Fig. S32 for details). Here $R_{\mathrm{m}}$ is the size of a SDS molecule, $c_0$ their concentration in units of mol l$^{-1}$, $N_{\mathrm{A}}$ the Avogadro constant and $S_{\mathrm{T}}^{\mathrm{m}}$ the Soret coefficient of SDS. For $R_{\mathrm{m}} = 2$ nm[45], $c_0 = 5$ mM and $S_{\mathrm{T}}^{\mathrm{m}} = 0.03$ K$^{-1}$ [46] we find $-0.43$ μm$^2$ K$^{-1}$ s$^{-1}$ for the additional depletion contribution, which exceeds the thermophoretic mobility, $D_{\mathrm{T}} \approx 0.3$ μm$^2$ K$^{-1}$ s$^{-1}$, rendering the overall mobility negative. The PS NPs and the AuNPs are thus driven to the heated Au film surface (Fig. 5f, Supplementary Video 10) which allows for further transport in the thermo-osmotic boundary flow. Additional contributions, for example, thermo-electric fields may even enhance the attractive components. Overall, this concept is readily transferred to other objects as shown in Fig. 5h and Supplementary Video 12, where

we have trapped ellipsoidal PS particles in a 5 mM solution of SDS. Note that as compared to other schemes, our approach always includes thermo-osmotic boundary flows.

Summarizing, we have demonstrated that thermo-hydrodynamic boundary flows can manipulate nano-objects with unprecedented flexibility in a very simple sample geometry. These flows are the key for future thermo-optofluidic implementations with an extensive range of applications in the fields of (i) nanoparticle sorting and separation; (ii) assembly of nanophotonic circuits[47] and plasmonic quantum sensors[4,5]; (iii) biotechnology on-chip laboratories[48] and (iv) manufacturing of nanomaterials[4,6] and functional nanosurfaces[49,50]. We have substantiated our experimental findings of thermo-osmotic flow-assisted trapping with a quantitative theoretical description. A flow field multi-plexing scheme has been further developed to allow for the simultaneous manipulation of many individual nano-objects and the generation of complex effective flow patterns. Our concept can be combined with other thermally induced effects such as thermophoresis, depletion forces and thermo-viscous flows to form a fully-featured nanofluidic system-on-a-chip. Besides direct consequences for the field of plasmonic nanotweezers and other thermo-plasmonic trapping schemes, the use of thermo-hydrodynamic flows as a tool for nanofluidic applications will extend the limits at the forefront of nanotechnology and help to develop AI and feedback-controlled schemes for material synthesis.

## Methods

**Experimental setup.** The experimental setup (Fig. 1b, Fig. S1) consists of an inverted microscope (Olympus, IX71) with a mounted piezo translation stage (Physik Instrumente, P-733.3). The microparticles are heated by a focused, continuous-wave laser at a wavelength of 532 nm (CNI, MGL-III-532). The beam diameter is increased by a beam expander and sent to an acousto-optic deflector (AA Opto-Electronic, DTSXY-400-532) and a lens system to steer the laser focus in the sample plane. The deflected beam is focused by an oil-immersion objective (Olympus, UPlanApo ×100/1.35, Oil, Iris, NA 0.5–1.35) to the sample plane ($w_0 \approx 0.8$ μm beam waist in the sample plane). The sample is illuminated with an oil-immersion darkfield condenser (Olympus, U-DCW, NA 1.2–1.4) and a white-light LED (Thorlabs, SOLIS-3C). The scattered light is imaged by the objective and a tube lens (250 mm) to an EMCCD (electron-multiplying charge-coupled device) camera (Andor, iXon DV885LC). The variable numerical aperture of the objective was set to a value below the minimum aperture of the darkfield condenser. The dichroic beam splitter (D) was selected to reflect the laser wavelength (Omega Optical, 560DRLP) and a notch filter (F) is used to block any remaining back reflections from the laser (Thorlabs, NF533-17). The acousto-optic deflector (AOD), as well as the piezo stage, are driven by an AD/DA (analog-digital/digital-analog) converter (Jäger Messtechnik, ADwin-Gold II). A LabVIEW program running on a desktop PC (Intel Core i7 2600 4 × 3.40 GHz CPU) is used to record and process the images as well as to control the AOD feedback *via* the AD/DA converter. Typically, images have been recorded with an inverse framerate of $\tau = 20$ ms.

**Sample preparation.** The sample consists of two glass coverslips ($22 \times 22$ mm) confining a thin liquid film. First, the coverslips were thoroughly cleaned in an ultrasonic bath with Hellmanex III (1%), acetone, isopropanol, and Milli-Q water followed by 3 min plasma cleaning (PDC-32G, Harrick Plasma Inc.). Then, one of the coverslips was coated with a 50 nm gold film using a thermal evaporator (UNIVEX 300, Leybold GmbH) and a 5 nm chrome adhesion layer. Subsequently, the edges of the uncoated coverslip were covered with a thin layer of PDMS for sealing. The particle solution used for the experiments was prepared by dispersing gold nanoparticles (Cytodiagnostics Inc.), PS particles (microParticles GmbH) in different solutions of NaCl and SDS (Sigma–Aldrich). Finally, 0.3 μl of the mixed particle suspension is pipetted in the middle of one of the coverslips and the other is placed on top. Depending on the area covered by the liquid, typically about $10 \times 10$ mm, the resulting liquid film height is about 3 μm.

**Zeta potential measurements.** The zeta potential of the particles has been estimated using electrophoretic light scattering (Malvern Instruments Ldt., Zetasizer Nano ZS).

**COMSOL simulations.** The numerical simulation presented in this study were computed using COMSOL Multiphysics 5.2 using the Heat Transfer in Fluids and the Non-Isothermal Laminar Flow interface. The corresponding geometry is

depicted in Fig. S13 in Supplementary Note 7 together with the used parameters as well as Supplementary Note 9, Fig. S21. Numerical simulation results are displayed in Figs. S14–S17 and Fig. S22, S23.

## Data availability

The datasets produced in this study are available from the corresponding author upon reasonable request.

## Code availability

The Python code for the single particle tracking software is available at the corresponding GitHub Repository. Python scripts used to analyze the datasets and the files for the COMSOL simulations are available at the GitHub repository for this paper.

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

## Acknowledgements

We acknowledge financial support by the German Research Foundation (Deutsche Forschungsgemeinschaft, DFG) through the Collaborative Research Center TRR 102 "Polymers under multiple constraints: restricted and controlled molecular order and mobility" (DFG project number 189853844, SFB TRR 102, to F.C.) and project CI 33/14-1 (DFG project number 242631004 to F.C.). This work was funded by the Federal Ministry for Economic Affairs and Energy based on a resolution of the German Bundestag (BMWi, STARK program, project number 46SKD023X to F.C.) and is cofinanced from tax revenues on the basis of the budget passed by the Saxon state parliament (SMWK). We thank Andrea Kramer for carefully reading the manuscript.

## Author contributions

M.F. and F.C. designed the experiments. M.F. performed the experiments. M.F. and F.C. analyzed the experimental data. M.F. and F.C. implemented and evaluated the numerical calculations. M.F. and F.C. wrote the manuscript. All authors discussed the results and commented on the manuscript.

## Funding

## Competing interests

The authors declare no competing interests.
