## [Peer Review File · Nature Communications]

REVIEWER COMMENTS

Reviewer #1 (Remarks to the Author):

The manuscript by Franzl et. al. describes how to harness the contribution of thermo-osmotic flow and the balance between attractive van der Waal's force and electrostatic repulsion to trap gold nanoparticles in a salt solution. The phenomenon is appreciable when trapping gold nanoparticles due to the weak contribution from repulsive thermophoresis, given the high thermal conductivity of gold. The manuscript is well-written and should be of broad interest to the readers of Nature Communications journal. I have provided some technical comments questions to be addressed below:

- 1.) Most of the SI videos show the trapping of a single gold NP. Is this trap self-limiting? Can more than one gold NP be trapped? Also, please provide information on the spatial width of the trapping potential well?
- 2.) Does the channel height have to be limited to 3um? Can larger channel heights such as 100 um be utilized?
- 3.) What is the impact of the channel height on the thermo-osmotic flow and the trapping potential described in this manuscript? Is there a specific reason for limiting the channel height to only 3 microns?
- 4.) What is the smallest particle size of gold nanoparticles that can be stably trapped?

Reviewer #3 (Remarks to the Author):

The authors demonstrate interesting manipulation (trapping and driving, with multiplexing) of nanoparticles in thin film fluids using local laser-induced heating. They provide experimental data on the influence of the nanoparticle nature (metallic or dielectric) and dissolved species (salt, surfactant...) which are then discussed and compared to simulations, in good agreement providing a comprehensive description of the involved processes. This work can be of great interest for the direct use of the demonstrated manipulation technique, and for future development based on the theoretical description. Yet, the manuscript shows several weak points, with missing information, as well as statements with no clear enough support. Therefore, I would support publication after significant modifications, following the remarks below.

P3:

"while at lower salt concentrations an enhanced probability (Fig. 2a) of finding the particle near the surface is the cause of the observed diffusion coefficient."

Isn't this effect occurring at $c_0=10$ mM? The particle is trapped near the surface by the potential, and experience there the lower diffusion coefficient.

At lower concentrations, the particle is almost delocalized over the full height of the film, only with a slight shift towards the surface with increasing c_0 inducing a small reduction of the diffusion coefficient. This sentence appears at least not clear enough.

Can $\langle d \rangle$ be measured experimentally from z position statistics? Histograms of positions as function of c_0 could be of interest to compare with model.

Deviation may be found since calculated $p(d)$ only takes into account the potential, and actual probability density would also be influenced by the dynamics: lower diffusion will increase the effective time of presence near the surface. Can this consideration explain the discrepancy between modeled and experimental diffusion coefficient (lower experimental D at $c_0=1$ & 3 mM compared to model in fig 2b)?

Fig 2i,k

There is a significant discrepancy between experiment and simulation for the velocity in the xy plane with r close to 0 and away from the Au surface. Can this be discussed? It is not done in main text, where experimental data are considered to "compare well to simulation results".

P4:

"Analysis of the lateral position histograms (inset in Fig. 3c for 1.25 mW) yields an effective stiffness of the trap (Fig. 3a) that well matches the predictions based on the thermo-osmotic flow (Fig. 2i, j)." Figures 2i,j present experimental data of velocities. It is not explained how based on these data, prediction for stiffness are made. In fact, how the experimental and simulated stiffness in Fig 3a are obtained is not presented. Although I do not doubt on the conclusions, the methods for extracting the stiffness (exp and sim) should be clearly described in SI.

SI section 7

FE modeling for temperature profile with a constant heating flux can show strong dependence with boundary conditions (position and temperature of reservoirs fully drive the temperatures). Is $DT=25K$ (significantly) dependent on the thickness of glass? Is $T_{amb} = 25^{\circ}C$ well fixed in experiment (and how)?

Also, are all the FE modeling conclusions sensitive (or not) to water thickness? It is only discussed for thermal convection. It is important since this value is not measured and not precisely controlled. It is noted in Sample preparation that it is around $5\mu m$, while $3\mu m$ is considered in all models and schematics...

Below can be found many remarks and suggestions (and I might have missed some). Their amount indicates that significant writing effort needs to be done.

Fig 1:

Value of R could have been indicated in the schematics.

The " d vs V " graph is not presented at all in caption. It shows a selection of data, not indicating which exactly, only discussed later in fig2a. It should be removed or well introduced and discussed.

Page 1

z should be defined when it is first introduced in equation ($d = z - R$). Definition should explicitly mention that it is the distance of the "center" of the particle from the wall.

c_0 is not defined in main text.

Fig 2:

a-b-c can be separated from the rest for clarity, as they correspond to a different section in the text.

d- Caption can mention that glass interfaces are presented as white solid lines at $z = 0$ and $3\mu m$.

Page 4

η ζ ϵ AH are not defined in main text.

Fig 3

b- F° instead of F^{OF} in the figure

shaded and dashed areas are not described in caption

e- The corresponding video could have been included as SI.

Page 6

Eq 6 and 7, please differentiate the drift velocity symbol " u " in the 2 cases.

Eq 7, R already corresponds to the AuNP size. c seems to be concentration of SDS (needs to be explicitly mentioned in main text), while c_0 is in the equation.

NA is not defined.

Please correct « the the »

Sample preparation:

Solution where Au particles are dispersed is not indicated

Financial supports are not implemented.

Author contributions:

"und" instead of and

SI:

figS14: issue with the curve-color attribution.

Response to Reviewer Comments

Reviewer #1 (Remarks to the Author):

Thank you very much for the valuable comments and questions!

1.) Most of the SI videos show the trapping of a single gold NP. Is this trap self-limiting? Can more than one gold NP be trapped? Also, please provide information on the spatial width of the trapping potential well?

The trap is not self-limiting, i.e., we can trap more than one gold NP at a single trapping location. To show this, we have added a new Supplementary Video (Video 13). The trapping potential is illustrated the inset in Fig. 3c and its spatial width (σ) in terms of the trapping stiffness $k = (k_B T)/\sigma^2$ is plotted as function of the heating power in Fig. 3a. We updated also the figure caption of Fig. 3 highlight this result more directly.

2.) Does the channel height have to be limited to 3 μm ? Can larger channel heights such as 100 μm be utilized?

There is no limitation to a specific channel height. Channel heights of several 100 μm can be utilized as well. For larger channel height additional convective contributions may arise, which are, however, weak as compared to the thermo-osmotic flows (see Fig. S26).

3.) What is the impact of the channel height on the thermo-osmotic flow and the trapping potential described in this manuscript? Is there a specific reason for limiting the channel height to only 3 microns?

The strength of thermo-osmotic flows at the gold/water interface is not affected by the channel height, but a larger sample height leads to weaker back flows in the center of the liquid film. The sample height of 3 μm was chosen to provide easy experimental access to the z-position of the particles (Fig. S2). At higher sample heights, the particles go out of focus due to the limited depth of field of the microscope objective and the particle intensities become too low to be detected.

4.) What is the smallest particle size of gold nanoparticles that can be stably trapped?

So far, we have achieved a stable trapping of 50 nm AuNPs (Fig. 4a). As the trapping is defined by the strength of the DLVO potential and the optical/hydrodynamic forces that try to push the particle out of the DLVO potential, optimal trapping conditions with different salt concentration can be found also for smaller particles (see "Concentration/Size Dependence" in Supplementary Note 4). Also, the use of depletion forces, as indicated in the manuscript, can further increase the range of particle sizes that can be trapped. Nevertheless, we have not tested the limits of trapping so far.

Reviewer #3 (Remarks to the Author):

Thank you very much for carefully evaluating our manuscript. We have revised our manuscript according to the questions and comments. Please find additional details below.

P3: "while at lower salt concentrations an enhanced probability (Fig. 2a) of finding the particle near the surface is the cause of the observed diffusion coefficient."

Isn't this effect occurring at $c_0=10$ mM? The particle is trapped near the surface by the potential, and experience there the lower diffusion coefficient.

At lower concentrations, the particle is almost delocalized over the full height of the film, only with a slight shift towards the surface with increasing c_0 inducing a small reduction of the diffusion coefficient. This sentence appears at least not clear enough.

Thank you for pointing out this weak point in the text. As the salt concentration increases, the particle spends more and more time near the surface due to the increasing depth of the DLVO potential. However, this increasing time near the surface leads to a strongly decreasing mean distance $\langle d \rangle$ only for a salt concentration $c_0 > 3$ mM as calculated. At about $c_0=10$ mM the mean distance approximately corresponds to the minimum of the DLVO potential, as shown in Figure 2c. This indicates a stable trapping. At the concentrations between 3 mM and 10 mM, the particle would be temporarily trapped for increasing periods of time. We have now reformulated this paragraph and updated also Figure 2c to make the overall message is clearer.

Can $\langle d \rangle$ be measured experimentally from z position statistics? Histograms of positions as function of c_0 could be of interest to compare with model. Deviation may be found since calculated $p(d)$ only takes into account the potential, and actual probability density would also be influenced by the dynamics: lower diffusion will increase the effective time of presence near the surface. Can this consideration explain the discrepancy between modeled and experimental diffusion coefficient (lower experimental $D||$ at $c_0=1$ & 3 mM compared to model in Fig. 2b)?

$\langle d \rangle$ can in principle be measured from the z -position, yet the statistics is distorted by non-linearities of the spot size as a function of z -position at small z . Close to the focus, the size of the spot converges nonlinearly to the focal point spread function, which translates into a higher probability density to measure small spot sizes and thus distances. This is less important for the velocity determination (due to the differences it just lowers the measured velocity value), but it strongly affects the mean distance measurement and we have not been able yet to correct for this issue.

Overall, the value of $\langle d \rangle$ is determined solely by the potential at thermal equilibrium. While diffusion near the interface is slowed down and the residence time is increased, the time required for diffusion to the interface is also longer. Thus, without additional potential, diffusion in thermal equilibrium will always result in a homogeneous density distribution. This is also the result of Eq. 10 in the Appendix when the potential is set to zero. In this case, the particle needs the same time for the path from d_{\min} to d_{\max} as for the path from d_{\max} to d_{\min} , which expresses the validity of the detailed balance.

However, the longer residence times due to slower diffusion make it experimentally more difficult to observe the correct mean distance in an experiment with limited measurement time. We assume that this limitation, together with the uncertainty in the sample height, causes the discrepancy between the theoretical and experimental diffusion coefficients.

Fig 2i,k: There is a significant discrepancy between experiment and simulation for the velocity in the xy plane with r close to 0 and away from the Au surface. Can this be discussed? It is not done in main text, where experimental data are considered to “compare well to simulation results”.

The discrepancy is caused by the limited spatial accuracy. Very small spatial distances ($r = 0$) are not well resolved and the particles at this position are very fast. For a particle speed of $60 \mu\text{m/s}$, this could be already more than $1 \mu\text{m}$, which makes a precise localization and velocity determination very difficult in this region. To improve this, we would need to go to much higher framerates. We added a sentence to discuss the discrepancy in the main text.

P4: “Analysis of the lateral position histograms (inset in Fig. 3c for 1.25 mW) yields an effective stiffness of the trap (Fig. 3a) that well matches the predictions based on the thermo-osmotic flow (Fig. 2i, j).”

Fig. 2i,j present experimental data of velocities. It is not explained how based on these data, prediction for stiffness are made. In fact, how the experimental and simulated stiffness in Fig 3a are obtained is not presented. Although I do not doubt on the conclusions, the methods for extracting the stiffness (exp and sim) should be clearly described in SI.

Thank you very much for the comment. As mentioned in the main text, the velocity fields are converted to hydrodynamic forces with $F = 6\pi \eta R \gamma v$. We have added a short explanation to the main text pointing also to a section in the SI, which explains additional details.

SI Section 7: FE modeling for temperature profile with a constant heating flux can show strong dependence with boundary conditions (position and temperature of reservoirs fully drive the temperatures). Is $\Delta T=25\text{K}$ (significantly) dependent on the thickness of glass? Is $T_{\text{amb}} = 25^\circ\text{C}$ well fixed in experiment (and how)? Also, are all the FE modeling conclusions sensitive (or not) to water thickness? It is only discussed for thermal convection. It is important since this value is not measured and not precisely controlled. It is noted in Sample preparation that it is around $5\mu\text{m}$, while $3 \mu\text{m}$ is considered in all models and schematics...

The limit affecting the thickness of the glass was chosen sufficiently large ($50 \mu\text{m}$), as indicated in Figure S21 in the Appendix. We find no significant change in the maximum temperature rise with the thickness of the glass. Very small water film thicknesses will reduce the maximum temperature rise due to the greater thermal conductivity of glass compared to water. We have added a paragraph to the SI to explain this further. The ambient temperature is set and measured with a foil heater and an integrated thermistor attached to the lens and controlled with a temperature controller.

The sample height specified in the “Sample preparation” section was incorrect. The samples were prepared with a nominal thickness of $3 \mu\text{m}$. Many thanks for indicating this flaw!

Below can be found many remarks and suggestions (and I might have missed some). Their amount indicates that significant writing effort needs to be done.

We thank the reviewer for carefully reading the manuscript. We addressed all mentioned issues.

Fig 1: Value of R could have been indicated in the schematics. The “ d vs V ” graph is not presented at all in caption. It shows a selection of data, not indicating which exactly, only discussed later in fig2a. It should be removed or well introduced and discussed.

This has been corrected.

Page 1: z should be defined when it is first introduced in equation ($d = z - R$). Definition should explicitly mention that it is the distance of the “center” of the particle from the wall. c_0 is not defined in main text.

This has been corrected.

Fig 2: a-b-c can be separated from the rest for clarity, as they correspond to a different section in the text. d- Caption can mention that glass interfaces are presented as white solid lines at $z = 0$ and $3 \mu\text{m}$.

We have split Figure 2 into two Figures and extended the figure captions to better reflect all details.

Page 4: η ζ ϵ AH are not defined in main text.

This has been corrected.

Fig 3: b- $F^{\wedge}O$ instead of $F^{\wedge}OF$ in the figure shaded and dashed areas are not described in caption

This has been corrected.

e- The corresponding video could have been included as SI.

This has been corrected.

Page 6: Eq 6 and 7, please differentiate the drift velocity symbol "u" in the 2 cases.

This has been corrected.

Eq 7, R already corresponds to the AuNP size.

This has been corrected.

c seems to be concentration of SDS (needs to be explicitly mentioned in main text), while c_0 is in the equation.

This has been corrected.

NA is not defined.

This has been corrected.

Please correct « the the »

This has been corrected.

Sample preparation: Solution where Au particles are dispersed is not indicated.

This has been corrected.

Financial supports are not implemented.

We have now indicated financial support.

Author contributions: "und" instead of and.

This has been corrected.

SI: FigS14: issue with the curve-color attribution.

We could not find this issue.

REVIEWERS' COMMENTS

Reviewer #1 (Remarks to the Author):

I thank the authors for addressing my comments. I recommend acceptance of the manuscript.

Reviewer #3 (Remarks to the Author):

Authors' response to my comments and corresponding modifications in the manuscript fully fulfill my expectation.

I therefore recommend publication of this work in Nature Communications.

Response to Reviewer Comments

Reviewer #1 (Remarks to the Author):

I thank the authors for addressing my comments. I recommend acceptance of the manuscript.

Thank you very much for your positive assessment and the time for reviewing our manuscript.

Reviewer #3 (Remarks to the Author):

Authors' response to my comments and corresponding modifications in the manuscript fully fulfill my expectation. I therefore recommend publication of this work in Nature Communications.

Thank you very much for your positive assessment and the time for reviewing our manuscript.